# Signaling Mechanisms of Stem Cell Therapy for Intervertebral Disc Degeneration

**DOI:** 10.3390/biomedicines11092467

**Published:** 2023-09-06

**Authors:** Xiaotian Du, Kejiong Liang, Shili Ding, Haifei Shi

**Affiliations:** Department of Orthopedic Surgery, The First Affiliated Hospital, Zhejiang University School of Medicine, Hangzhou 310003, China; 21618580@zju.edu.cn (X.D.); lkj1020@zju.edu.cn (K.L.); 1515071@zju.edu.cn (S.D.)

**Keywords:** intervertebral disc, tissue degeneration, signaling pathway, stem cell treatment

## Abstract

Low back pain is the leading cause of disability worldwide. Intervertebral disc degeneration (IDD) is the primary clinical risk factor for low back pain and the pathological cause of disc herniation, spinal stenosis, and spinal deformity. A possible approach to improve the clinical practice of IDD-related diseases is to incorporate biomarkers in diagnosis, therapeutic intervention, and prognosis prediction. IDD pathology is still unclear. Regarding molecular mechanisms, cellular signaling pathways constitute a complex network of signaling pathways that coordinate cell survival, proliferation, differentiation, and metabolism. Recently, stem cells have shown great potential in clinical applications for IDD. In this review, the roles of multiple signaling pathways and related stem cell treatment in IDD are summarized and described. This review seeks to investigate the mechanisms and potential therapeutic effects of stem cells in IDD and identify new therapeutic treatments for IDD-related disorders.

## 1. Background

Low back pain is the leading global disability [1]. To date, intervertebral disc degeneration (IDD) has become the primary clinical risk factor for low back pain and the pathological basis for developing disc herniation, spinal stenosis, and spinal deformities [2]. It is reported that ordinary populations have a 10% lifetime prevalence of sciatica-related low back pain [3]. While a number of approaches are used to treat symptomatic IDD-related diseases, there are marked heterogeneities in therapeutic efficacies. For instance, surgery is indicated for disc herniation patients who failed conservative treatments, but back pain and leg pain remained in approximately a third of surgical cases two years later [4]. Such heterogeneities in clinical outcomes reflect the need for early diagnosis and precise prognostic judgment.

Anatomically, the intervertebral disc (IVD) connects vertebral bodies in the spine with three compartments: nucleus pulposus (NP), annulus fibrosus (AF), and cartilaginous endplate (CEP). IDD causes decreased water content of the NP and AF, loss of elasticity of the NP, centripetal fissures, structural changes of collagen fibers in the AF, extensive damage in the CEP, subchondral osteosclerosis, angiogenesis, neoinnervation, significant reduction or even loss of IVD height, and IVD-related biomechanical changes. Degenerated IVD cells have fewer active cells, aberrant extracellular matrix metabolism, and pro-inflammatory chemicals [5].

To date, IDD pathology is unclear. Mechanical stress, trauma, infection, genetic vulnerability, and inflammation can increase IDD pathology [6]. Recent developments in gene microarray technology have yielded fresh insights into the molecular pathogenesis of IDD-related diseases. Using single-Cell RNA Sequencing technology, several cell types including chondrocyte 1–5, endothelial, macrophage, neutrophil, and T cells were delineated in IVD. Specifically, chondrocytes 5 expressing FN1, SESN2, and GDF15, and chondrocytes 4 expressing PTGES, TREM1, and TIMP1 may exacerbate IDD, while chondrocytes 2 expressing MGP, MT1G, and GPX3 may mitigate this degenerative process [7]. Regarding molecular mechanisms, cellular signaling pathways such as Wnt/β-catenin, NF-κB, mitogen-activated protein kinase (MAPK), lipoyl inositol-3 kinase (PI3K)/serine-threonine protein kinase (Akt), and transforming growth factor β (TGF-β)/Smads constituted a complex network of signaling pathways that coordinate the cell survival, proliferation, differentiation, and metabolism. Studying the molecular pathogenesis of IDD and delaying or correcting its pathological alterations is a key problem and research hotspot in orthopedics.

Stem cells are multipotent, self-renewing cells, and are implicated in various basic processes, such as cellular differentiation, proliferation, angiogenesis, oxidative stress response, inflammation, and extracellular matrix synthesis [8]. The potential of stem cell therapy has been investigated in the treatment of degenerative musculoskeletal diseases [9]. Recently, stem cells derived from NP, CEP, bone marrow, and adipose tissue have shown great potential in clinical applications for IDD by regulating signaling pathways in the IDD process. The present review was made to investigate the mechanisms and potential therapeutic effects of stem cells in IDD and identify new therapeutic treatments for IDD-related disorders. Recent advances in IDD-related signaling pathways and related stem cell treatment in IDD are summarized and described below.

## 2. Wnt/β-Catenin Signaling Pathway

The classical Wnt signaling pathway includes secreted Wnt family proteins, transmembrane receptor proteins of the Frizzled family (Dishevelled, GSK3, Axin, APC, and β-catenin), and downstream transcriptional regulators of the TCF/LEF family. This route involves embryonic development, stem cell proliferation, and degenerative disorder development [10]. For the skeletal system, the Wnt signaling pathway was crucial for developing craniofacial, limb, and joint structures, and mutations in members of this pathway would lead to skeletal malformations in mice and humans [11].

The Wnt signaling pathway’s dynamic activity during IVD growth, maturation, and degeneration has been studied (Table 1). Excessive activation of this pathway, for example, may lead to severe structural malformations in IVD, as evidenced by disruption of the growth plate, excessive cellular proliferation, disruption of the lamellar structure in the AF, and reduction in proteoglycans in the NP. β-catenin deficiency also accelerates bone formation between the CEP and growth plate [12]. Moreover, for the degenerative process, the Wnt/β-catenin signaling pathway activation can accelerate this process by inducing the inflammatory factors production [10], promoting cellular apoptosis and senescence [13], and degradation of the extracellular matrix of IVD cells [14]. For example, conditional activation of β-catenin in mice can lead to severe structural defects in IVD [15]. Furthermore, the upregulation of β-catenin in the canine IVD can upregulate the Runx2 expression in the IVD and promote degenerative calcification in the IVD [14]. Additionally, in IVD, WNT/β-catenin pathway activation promotes cellular senescence, matrix disintegration, and IDD [13].

Various active substances can promote IDD by upregulating the Wnt/β-catenin pathway expression (Table 1). For example, lncRNA HOTAIR and circITCH can promote cellular senescence, apoptosis, and matrix degradation in IVD by activating the Wnt/β-catenin pathway [16]. In IVD cells, TNF-α and Wnt signaling can generate a positive feedback loop [17]. IDD may be alleviated by inhibiting this mechanism. For example, RBMS3 RBMS3 (RNA binding motif, single-stranded interacting protein 3) is a member of the c-myc single-strand binding protein family and encodes an RNA-binding protein [18]. In addition, by inhibiting the Wnt/β-catenin signaling pathway, RBMS3 can enhance the proliferative capacity of IVD cells and suppress apoptosis and inflammatory responses in IVD [19].

**Table 1 biomedicines-11-02467-t001:** Effects of signaling pathway activation for IDD and pathway activator.

Signaling Pathway	Wnt/β-Catenin Signaling Pathway	NF-κB Signaling Pathway	MAPK Signaling Pathway	PI3K/Akt Signaling Pathway	TGF-β1 Signaling Pathway
Effects of pathway activation for IDD	↑[10,13,14,15]	↑[20,21,22,23,24,25,26,27,28,29]	See details in Table 2	↓[30,31,32,33,34,35,36,37,38]	↑[39,40,41] ↓[42,43,44,45,46,47,48,49,50,51]
Activator	LncRNA HOTAIR [16], circRNA ITCH [52], TNF-α [17]	TREM2 [53], CGRP [28], Ca^2+^ [24], IL-1β [25], HMGB1 [20], N-Ac-PGP [21], ROS [22], S100A9 [26], ARG2 [27]	CHI3L1 [54], ROS [22,55], MALAT1 [56], Resistin [57], Syndecan-4 [58], IL-17A [59], IAPP [60], Glucose [61], Visfatin [62]	17Beta-estradiol [34], BMP2 [33], Apelin-13/APJ [35], Resveratrol [63]	Smad3 [43], ASIC3 [42], caveolin-1 [46], Parathyroid hormone [50]

↑: Deteriorating effect. ↓: Mitigating effect.

As mentioned, the Wnt/β-catenin signaling pathway plays a crucial role in IDD and may function as a potential therapeutic target for stem-cell-related treatment. For example, the aberrant apoptosis of NP cells is one of the most remarkable pathological changes in IDD development. The compression leads to an increase in apoptosis and Wnt-related gene expression, which can both be suppressed by the in vitro co-cultured mesenchymal stem cell (MSC) [64]. Moreover, the age-related variation of Wnt signaling in IVD cells may limit regeneration by depleting the progenitors and attenuating the expansion of chondrocyte-like cells [65]. During IDD, CEP gradually calcified and the osteogenic differentiation was increased [66]. Cartilage endplate stem cells (CESCs) are essential for IDD by regulating chondrogenesis and osteogenesis in the CEP [67]. Downregulation of WNT5A was proved to inhibit IDD via downregulating the osteogenic differentiation of CESCs [68]. Exosomes derived from CESCs, however, can activate HIF-1α/Wnt signaling via autocrine mechanisms to increase the expression of GATA4 and TGF-β1, thereby promoting the migration of CESCs into the IVD and the transformation of CESCs into NP cells and inhibiting IDD [69]. Therefore, the activation of the Wnt signaling pathway in IVD stem cells may also reveal its alleviating effects in IDD. For example, the Wnt/-catenin pathway in IVD can be activated by bone marrow mesenchymal stem cells (BMSCs)-derived extracellular vesicles, leading to the suppression of cellular apoptosis, ECM degradation, and IDD progression [70]. Notably, the overexpression of Wnt11 in adipose-derived stem cells (ADSCs) induces the ADSCs cells differentiating to the NP cells, which may have a potential utility for the treatment of IDD [71] (Figure 1).

## 3. NF-κB Signaling Pathway

NF-κB protein, initially found in B lymphocyte extracts, binds to enhancer regions of immunoglobulin light chain genes [72]. In the classical NF-κB signaling pathway, IκB kinase (IKK) regulated the IκB proteins’ phosphorylation [73]. For IVD, NF-κB nuclear translocation upregulation accelerates IDD [29]. For example, HMGB1, a pro-inflammatory factor, upregulates the NF-κB signaling pathway in IVD cells to induce inflammatory cytokines and matrix metalloproteinases [20]. Additionally, in degenerative IVD, the neuropeptide CGRP and its receptors are overexpressed, which inhibits cellular growth and promotes apoptosis and inflammation by upregulating the NF-κB signaling pathway [28]. Notably, inflammatory mediators and chemokines produced by the NF-κB signaling pathway activation formed a vicious cycle in the IDD process [23]. For example, the NF-κB pathway activation by IL-1β would also promote the IL-1β precursors expression, accelerating IVD degeneration [24]. Another study showed that IL-1β could also regulate the miR-133a-5p/FBXO6 axis expression through the NF-κB pathway, which would regulate the proliferation of IVD cells and apoptosis [25].

Besides regulating the inflammatory responses [25], the NF-κB signaling pathway upregulation can also deteriorate IDD by promoting matrix metalloproteinases and destructing the cellular matrix of IVD [20]. For instance, the inflammatory chemokine N-Ac-PGP promotes NF-κB and MAPK signaling pathways in NP cells to generate pro-inflammatory cytokines and matrix catabolic enzymes [21]. Moreover, the increase in neovascularization in aging IVD would exacerbate the oxidative stress for this tissue. Upregulation of reactive oxygen species (ROS) would induce catabolic and inflammatory expression in IVD cells by stimulating the NF-κB pathway [22]. Moreover, the oxygen-sensing proteins would induce apoptosis, matrix degradation, and the inflammatory response for NP cells by NF-κB signaling pathway activation [26]. NF-κB can enhance oxidative stress, generating another vicious cycle between IDD and the oxidative stress [27]. Thus, NF-κB signaling pathway activation promotes IVD apoptosis, inflammatory response, matrix breakdown, and oxidative stress, which worsens IDD.

Studies have revealed the use of BMSCs in tissue-engineering treatments to slow or reverse IDD. The coculturing of BMSCs with disc-native NP cells promotes the matrix production of NP cells and the differentiation of BMSCs into NP-like cells through downregulating NF-κB pathway [74]. Moreover, TNF-α-stimulated gene 6 secreted by BMSCs can attenuate inflammation factors production, matrix degeneration, and IDD by inhibiting the NF-κB signaling pathway [75]. Interestingly, inflammation factors also revealed positive roles for stem cells in recent degenerative disease studies. Tumor necrosis factor-α (TNF-α) is critical for accelerating IDD. While with a relatively low concentration (0.1–10 ng/mL), TNF-α promotes the proliferation and migration of NP mesenchymal stem cells (NPMSCs) but inhibits their differentiation toward NP cells. Moreover, the NF-κB signaling pathway is activated during the TNF-α-inhibited differentiation of NPMSCs, and the NF-κB signal inhibitor can partially counteract the adverse effect of TNF-α on the differentiation of NPMSCs [76]. Moreover, TGF-β1 is a strong immune suppressor, whose increase would inhibit IκB phosphorylation and NF-κB activation. Co-culturing of NP cells with BMSCs significantly increases TGF-β1 in NP, leading to anti-inflammatory effects via the inhibition of NF-κB, and ameliorating IDD due to increased collagen II and aggrecan in the degenerative disc [77]. Cellular senescence is another promotive factor for IDD. Upon TNF-α stimulation, NF-κB activation reveals pro-senescence effects in NP cells, while co-culturing with BMSCs reduces senescence-associated β-galactosidase, matrix metalloproteinase 9, and NF-κB signaling in senescent NP cells. Accordingly, Zinc metallopeptidase STE24, whose dysfunction is related to premature cell senescence and aging, is restored upon BMSC co-culture and inhibits the effects of NF-κB activation [78]. Moreover, ataxia-telangiectasia mutated kinase is a vital component for NF-κB-mediated cellular senescence, stem cell dysfunction, and aging. Inhibition of this kinase also reduces activation of NF-κB, improves the functions of muscle-derived stem/progenitor cells, and thus alleviates IDD [79].

## 4. MAPK Signaling Pathway

The mitogen-activated protein kinase (MAPK) cascade signaling pathway has three main sub-pathways: the extracellular signal-regulated kinase 1/2 (ERK1/2) pathway, the p38 kinase pathway, and the c-Jun amino-terminal kinase (JNK1–3) pathway. All three sub-pathways involved physiological and pathological processes such as cell proliferation, differentiation, apoptosis, stress, and inflammatory responses (Table 2). As mentioned in Figure 1, the promotion of inflammation, oxidative stress, senescence, and death processes deteriorate IDD, while the activations of stem cell differentiation, proliferation of physiological cells, phenotype maintenance, and matrix maintenance mitigate this pathological process.

### 4.1. ERK1/2 Signaling Pathway

The MAPK/ERK pathway activation in AF and NP may have different or opposite roles [103]. MAPK/ERK pathway activation in AF helps IVD maintain its physiological phenotype, repair damage, and prevent tissue degeneration. For example, low-intensity pulsed ultrasound would enhance cell proliferation and collagen synthesis processes by activating the ERK pathway in AF, promoting the AF’s repair and alleviating IDD [84]. Additionally, the ERK pathway activation can significantly enhance the proliferation and migration of AF cells, promoting IVD repair [82,83,104]. Moreover, ERK maintains IVD function in acidic and hyperosmotic microenvironments [87] and the activation of this pathway would also activate AF cell regeneration in 3D culture [81]. For phenotypic maintenance in AF cells, however, the activated MAPK-ERK pathway revealed opposite roles in studies [86,88]. In NP tissue, MAPK/ERK pathway activation was linked to extracellular matrix breakdown, cellular senescence, apoptosis, inflammation, autophagy, and oxidative stress, worsening IDD pathology [105]. For example, the M1-type [80] and M2a-type [54] macrophages would promote the imbalance of extracellular matrix metabolism in NP cells by activating the ERK signaling pathway. Additionally, elevated oxygen tension-induced ROS in NP causes cell cycle arrest and senescence through ERK signaling pathway activation [22].

Both NPMSC and ADSC are used as candidate cells for IDD treatment. The ERK pathway is activated by the hyperosmolarity in the disc, which inhibits proliferation and chondrogenic differentiation of NPMSCs [106]. In another study, however, the activation of the MAPK/ERK signaling pathway leads to the enhancement of NPMSC viability, differentiation towards NP cells, and extracellular matrix biosynthesis in the disc [107]. Similarly, lithium, a common anti-depression drug, was found to promote ROS and ERK1/2 pathway, which enhances ADSC’s survival and ECM deposits in the degenerative disc [108]. Recently, scaffolds for IDD tissue engineering were designed for the maintenance of stem cells in the acidic environment of the disc. For example, Sa12b-modified hydrogel enhances the biological activity of NPMSCs by inhibiting acid-sensing ion channels by inhibiting the ERK signaling pathway [109]. In addition, collagen type II hydrogel significantly promotes extracellular matrix synthesis by activating the ERK pathway [110].

### 4.2. p38-MAPK Signaling Pathway

The p38-MAPK signaling pathway also regulated inflammation, cellular stress, growth and development, and apoptosis in IVD. Growth factors, inflammatory cytokines, and environmental stresses trigger IVD’s p38-MAPK signaling pathway, releasing inflammatory substances, and degrading the cellular matrix, thus accelerating IDD [111]. For example, non-physiological loading can stimulate apoptotic body production in AF cells by activating the p38-MAPK pathway, ultimately leading to the apoptosis and degeneration of IVD [96]. For chondrocytes in CEP, the p38-MAPK signaling pathway would also induce cellular apoptosis [56]. In recent years, the roles of resistin and endoplasmic reticulum stress have been revealed in multiple degenerative diseases. In IVD, these two variables activated the p38-MAPK pathway to produce pro-inflammatory effects [57,89].

Various research has proven the therapeutic effects of inhibiting the p38-MAPK pathway on IDD in recent years. All of the pulsed electromagnetic fields [90], tyrosine kinase inhibitors [91], and tanshinone IIA sulfonate [94] exert their anti-inflammatory activities for IVD cells by downregulating the p38-MAPK signaling pathway. Moreover, blocking the p38-MAPK pathway can greatly reduce the inflammatory consequences of non-physiological stress on IVD cells [97]. Moreover, the p38-MAPK pathway inhibition would protect NP cells against oxidative stress and mitochondrial dysfunction [95], prevent NP cells apoptosis by inhibiting M1-type macrophage polarization and promoting the release of anti-inflammatory factors from M2-type macrophages [93], and also increase the expression of IVD protective factors [98]. Additionally, ERK5 is another member of the MAPK family and regulates the maintenance of the extracellular matrix in IVD, and the suppression of ERK5 resulted in decreased type II collagen and aggrecan in NP cells, indicating the potential protective roles of MAPK family members in IDD [112].

In the disc, BMSC-derived extracellular vesicles have the potential to alleviate extracellular matrix degradation, apoptosis, and cell cycle arrest in IDD via downregulating phosphorylated p38 MAPK levels [93,113]. In addition, the suppression of p38 MAPK signaling with specific inhibitors also promotes the anti-inflammatory impact of MSCs and the alleviation of IDD [114]. The activation of the p38 signaling pathway, however, has also revealed its therapeutic potential for IDD by stimulating the differentiation of MSC in the disc. For example, TGF-β1 promotes the differentiation of MSC to NP-like cells in the disc’s physiological hypoxia environment by activating ERK and p38 signaling pathways [115]. Notably, the therapeutic effect of intervertebral fusion for IDD is still unsatisfactory and the conditioned medium of BMSCs treated with electromagnetic fields can promote osteogenic differentiation of BMSCs by activating the p38 signaling pathway, which accelerates intervertebral fusion for IDD treatment [116,117].

### 4.3. JNK Signaling Pathway

In IVD, JNK activation causes inflammation and matrix breakdown [58]. For example, aberrant expression of pancreatic amyloid polypeptide would increase the secretion of IL-1, TNF-α, and matrix-degrading enzymes in IVD by activating this pathway [60]. Similarly, IL-17A can exert a pro-inflammatory effect by stimulating the p38 and JNK pathways, causing NP cells to produce more COX2/PGE2 [59]. Recently, the endocrine function of adipose tissue was revealed. Visfatin, a protein secreted by adipose tissue, can induce IL-6 expression in NP cells by activating the JNK/ERK/p38-MAPK signaling pathway, thus promoting the inflammatory response and extracellular matrix degradation in IVD [62].

Increased JNK signaling pathway also upregulates IVD cell autophagy and apoptosis. Under mechanical stress stimulation, elevated ROS in rat NP cells activates the JNK signaling pathway and induces autophagy, thus accelerating IDD [55]. Moreover, IDD was also more common among people with diabetes than non-diabetics [118]. High glucose can lead to premature senescence of AF cells in young rats [101] and promote apoptosis of AF cells in a glucose concentration-dependent manner through activation of the JNK pathway [61]. Notably, JNK pathway suppression may also alleviate IDD. Crocin, the bioactive component of saffron, can alleviate the inflammatory and catabolic processes in IVD by JNK phosphorylation inhibition in NP cells [99]. Moreover, hinokitiol can also maintain the function of iron transport proteins and alleviate oxidative stress in NP cells by regulating the JNK pathway [100].

Inhibition of the JNK signaling pathway alleviates degeneration of stem cells derived from CEP, NP, and bone marrow [119]. For example, oxidative stress during the transplant of BMSC to degenerative discs may cause cell toxicity and poor survival of BMSCs. Mitophagy can maintain cellular homeostasis and defend against oxidative stress by eliminating dysfunctional or damaged mitochondria. Mechanically, oxidative stress facilitates mitophagy through the JNK signaling pathway at an early stage of IDD but decreases mitophagy and increases apoptosis at a late stage [120]. Moreover, excessive oxidative stress also induces apoptosis and senescence of NP stem cells. Heat shock protein 70 (HSP70), a cytoprotective and antioxidative protein, reveals its protective roles against apoptosis and senescence of NP stem cells by downregulating the JNK signaling pathway [121].

## 5. PI3K/Akt Signaling Pathway

PI3K/Akt also regulates cell survival, metabolism, and proliferation in numerous tissues [122]. IVD cells need PI3K/Akt pathway activation to survive hypoxic conditions [123]. A possible explanation for this role was proposed as the PI3K/Akt pathway activation would promote autophagy and inhibit apoptosis of NP-derived [38] and endplate-derived [36] stem cells, which protected IVD from oxidative damage and facilitated the repair of degenerative injury.

Moreover, PI3K/AKT signaling protected matrix production in NP cells, while inhibiting PI3K activity would decrease proteoglycans in the IVD matrix [30]. Specifically, the PI3K/Akt/FOXO3 signaling pathway activation would downregulate the MMP-3 expression and upregulate type II collagen and ACAN in NP cells [34]. Activating PI3K/AKT signaling reduces matrix breakdown and inflammation [32]. For example, PI3K/Akt signaling pathway activation by BMP2 [33] and the Apelin-13/APJ system [35] can not only promote the production of type II collagen, ACAN, SOX9, and downregulate matrix-degrading enzymes in IVD, but also significantly inhibit the inflammatory response and apoptosis of NP cells. As the key driver of the inflammatory cascade in IVD, IL-1β promotes NP cell death, inflammatory responses, extracellular matrix remodeling, endoplasmic reticulum stress responses, and mitochondrial dysfunction. The PI3K/Akt pathway inhibits these IDD-related activities [31,37].

Recently, drugs and physiotherapeutic means to alleviate the IDD process by modulating PI3K/Akt pathway activity have also emerged. As mentioned, high oxidative stress in NP cells would promote degenerative changes by increasing intracellular ROS production. While resveratrol can inhibit oxidative stress-related effects by PI3K/Akt pathway activation in NP cells [63]. For physiotherapeutic aspects, circulating mechanical traction [124] and low-intensity pulsed ultrasound [125] can also alleviate degenerative changes in the NP extracellular matrix by activating the PI3K/Akt pathway. For AF cells, PI3K/AKT signaling pathway activation would also alleviate the degenerative processes. For example, the activation of this pathway inhibits AF cell cadmium-induced apoptosis [126]. However, a recent study also revealed the promotive effects of the PI3K/AKT signaling pathway for angiogenesis in IVD [127]. Thus, the data suggest that PI3K/AKT signaling pathway activation may treat IDD.

Based on stem cell studies, promising tools and insights for PI3K/AKT pathway-related IDD therapeutics were offered in recent studies. Mechanically, disc-derived stem cells regulate the function of the disc by delivering exosomes. The CESC-derived exosomes inhibit apoptosis of NP cells and attenuated IDD in rats via activation of the PI3K/AKT pathway. Additionally, exosomes from normal CESC inhibit NP apoptosis and alleviate IDD more effectively than exosomes from degenerative CESC [38]. Moreover, CESCs overexpressing Sphk2-engineered exosomes activates the PI3K/p-AKT pathway as well as the intracellular autophagy of NP cells, which ultimately ameliorates IDD by balancing autophagy/senescence [128]. In addition, for NP progenitor cells (NPPCs), exosomes secreted by NPPCs derived from degenerative discs would even exacerbate AF degeneration by blocking the activation of the PI3K-Akt pathway [129]. Notably, NPPCs remain difficult to maintain in culture. Fibroblast growth factor (FGF) 2 and chimeric FGF, however, were reported to enhance the phenotype maintenance of NPPCs via PI3K/Akt and MEK/ERK signals [130]. In addition, 1,25(OH)_2_D_3_ can also attenuate oxidative stress-induced apoptosis and mitochondrial dysfunction to NPPCs through PI3K/Akt pathway [131].

MSCs can also attenuate IDD by regulating cellular mechanical properties and apoptosis in the disc. For example, co-culture of degenerative NP cells with MSCs resulted in significantly decreased mechanical moduli and increased biological activity in degenerative NP by activating AKT signaling [132]. In addition, MSC-derived exosomes can prevent NP cells from TNF-α induced apoptosis and alleviate IDD by targeting phosphatase and tensin homolog by activating the PI3K-Akt pathway [133]. Through the AKT and ERK signaling pathways, exosomes from urine-derived stem cells can significantly inhibit endoplasmic reticulum (ER) stress-induced apoptosis and IDD under pressure conditions [134]. Similarly, exosomes from BMSCs can attenuate ER stress-induced apoptosis in degenerative discs by activating AKT and ERK signaling [135].

## 6. Hedgehog Signaling Pathway

Hedgehog signaling regulates skeletal development and repair [136]. Hedgehog proteins regulate IVD maturation, degradation, and calcification [137]. Hedgehog is highly expressed in young and healthy IVD cells, diminishes with notochord cell phenotypic loss, and increases again in late IDD [138]. Hedgehog contains three homologous proteins: Sonic hedgehog (Shh), Indian hedgehog (Ihh), and Desert hedgehog (Dhh). Among them, Shh and Ihh are closely related to the IDD process as described as follows.

### 6.1. Shh Signaling Pathway

IVD development and function require an appropriate Shh signaling pathway expression [139] and the deficiency of this pathway has been proven to be related to the aging phenotype of NP cells [140,141]. During the embryonic stage, the notochord eventually undergoes segmentation and forms IVD, and a notochord sheath must wrap it to retain its usual rod-shaped structure. The Shh signal loss in early embryonic stages would lead to structural abnormalities in the notochord sheath, leading to aberrant development of IVD and vertebrae [142,143].

Shh signaling influenced IVD growth and differentiation after birth. Without this signaling pathway, NP cells would lose their reticular network and collapse into IVD’s core region, while AF cells would lose their polar layered structure. Mechanistically, blocking the Shh signal would lead to the downregulation of TGF-β signaling and the upregulation of BMP and Wnt signaling expression [140]. The IVD between the sacral vertebrae collapses and merges during childhood, forming a typical sacral structure. In addition, the collapse of the sacral IVD has been associated with the downregulation of Shh signaling in the NP cells. Conversely, Shh signaling activation in NP cells would reactivate dormant NP cells and initiate IVD regeneration [144].

The activation of the Shh signaling pathway was proved to facilitate the differentiation of pluripotent stem cells to notochordal cells [145]. As mentioned, ADSC-based therapy is a promising treatment for IDD, while the difficulty in inducing NP-like differentiation limits its applications. Collagen type II promotes ADSC proliferation and differentiation toward an NP-like phenotype through the activation of the Shh signaling pathway [146] while the Shh signaling pathway inhibitor reduces the NP-like differentiation from ADSCs [147]. Similarly, the histone demethylase KDM4B also promotes the osmolarity-induced NP-like differentiation of ADSC by activating Shh signaling [148].

### 6.2. Ihh Signaling Pathway

The Ihh gene was first expressed in mesenchymal cells and chondrocytes of limbs. Ihh expression is confined to hypertrophic chondrocytes during skeletal growth plate development. Ihh inhibits chondrocyte maturation during long bone growth, and its dysregulation prevents proliferating chondrocytes from hypertrophic differentiation [149]. For example, mice carrying null mutations of the Ihh gene exhibit severe destruction of the growth plate at the embryonic stage with abnormalities in the proliferation and maturation of chondrocytes [150]. Additionally, conditional knockout of Ihh leads to reduced proliferation of chondroprogenitor cells and chondrocytes and the pathological processes in chondrocytes, including apoptosis, ectopic hypertrophy, and subchondral bone degeneration [151].

Moreover, blood vessels’ premature infiltration, loss of normal columnar structure in growth plates, and ectopic hypertrophic chondrocyte formation were also revealed in neonatal Ihh-knockout mice. Then, after birth, Ihh knockout mice would exhibit disruption of the articular surface of long bones and premature fusion of growth plates, leading to dwarfism in the mice [152]. However, Ihh signaling also promotes chondrocyte development, according to research. For instance, Ihh-regulated parathyroid hormone-related protein (PTHrP) prevents premature growth plate cartilage hypertrophic differentiation. Meanwhile, Ihh can also stimulate the differentiation of periarticular chondrocytes to columnar chondrocytes through a PTHrP-independent pathway [153].

IVD research discovered that Ihh is significantly expressed in embryonic vertebrae endplate cartilage and chondrocytes [154]. Ihh pathway overexpression decreased chondrocytes and alterations in IVD extracellular matrix proteins. For example, upregulation of this pathway would promote the calcification in endplate cartilage and the degradation in the extracellular matrix, and inhibiting this pathway would reverse these degenerative processes [155]. In the NP, ROS would enhance Ihh expression and induce cellular apoptosis, and inhibiting the p-eIF2α/ATF4/Ihh signaling cascade axis reduces antioxidant enzyme degradation, ROS, and NP cell death [156]. Furthermore, microtubule-based cilia were found to be involved in regulating the developmental and degenerative processes of IVD. During IDD, the downregulation of intraflagellar transport protein 80 disrupts the transduction of the Ihh signaling pathway, resulting in apoptosis and disordered cellular proliferation and differentiation in IVD cells [157].

## 7. TGF-β Signaling Pathway

TGF-β1 is a ubiquitous growth factor that regulates various cells’ proliferation, migration, differentiation, and survival. In skeletal tissues, TGF-β1 was proven to regulate osteochondral development and maintenance by affecting metabolism in cartilage and bone [158]. Notably, the TGF-β signaling pathway is critical for IVD growth and preserves IVD tissues by increasing matrix formation, limiting matrix disintegration, and reducing inflammatory responses [51]. For example, morphological deformities, including spinal kyphosis, the decreased height of endplate chondrocytes, and disordered arrangement, were revealed in Smad3 knockout mice. At the molecular expression level, the IVD in these mice exhibited a decrease in type II collagen, TGF-β1, and proteoglycan. These results also suggested a positive role of TGF-β1 in alleviating IDD [43]. Furthermore, TGF-β signaling also helps spine development during embryogenesis and IVD growth and maintenance after birth [159].

By generating glycosaminoglycan, NP cells preserve the matrix’s water-binding capabilities, and activating the TGF-β-Smad3 axis would increase the synthesis of glycosaminoglycan in NP cells, thus maintaining the water content and organizational structure of IVD [45]. In inflammatory response regulation, TGF-β1 can act synergistically with the inflammatory factor inhibitor ML264 to alleviate the IL-1β-induced inflammatory response and matrix degradation in the NP tissue [49]. CCN2 is another matrix protein that has anti-inflammatory and homeostatic properties. In addition, TGF-β1 can induce CCN2 expression by activating Smad3 and AP-1 signaling pathways in NP cells, thus alleviating the IDD process [44]. TGF-β1 can also inhibit the pro-inflammatory factors expression, thus providing matrix protection and altering the NP cells’ overall secretory phenotype [47].

For functional maintenance and damage repair, the TGF-β/SMAD signaling pathway can regulate the miR-455-5p/RUNX2 axis to prevent mechanically induced endplate chondrocyte degeneration [48]. Moreover, inflammation or degenerative stimulation would cause the increase in TGF-β1, which can down-regulate the expression of sodium channel proteins and thus stabilize the Na^+^ flux and the proteoglycan metabolism of NP cells [42]. Similarly, scaffold protein caveolin-1 can promote IVD repair by enhancing TGF-β signal transduction [46]. Moreover, the parathyroid hormone can also activate the TGF-β/CCN2 signaling pathway expression in NP cells and maintain the height and homeostasis of IVD by enhancing the TGF-β1 activity and upregulating the ACAN level [50]. Contrarily, TGF-β1 upregulation would deteriorate the process of IDD, and inhibition of overexpressed TGF-β1 in degenerative IVD would promote the proliferation of NP cells and inhibit cellular senescence and apoptosis [41]. Regarding the cellular matrix, TGF-β1 can exacerbate the inflammatory and fibrotic manifestations of degenerating IVD [40]. Furthermore, the increased TGF-β1 activity can also increase the osmotic pressure of the extracellular environment and lead to IDD advancement [39].

The activation of the TGF-β signaling pathway can also alleviate IDD by increasing the differentiation of stem cells to NP-like cells [160]. For example, TGF-β pathway stimulation is a vital step in a protocol for directed in vitro differentiation of human pluripotent stem cells into notochord-like and NP-like cells of the disc [161]. TGF-β1 can also differentiate human ADSCs into NP cells, providing a new mechanism for its IDD-relieving effects [162]. Moreover, TGF-β signaling is also related to the homeostasis of cellularity and cellular matrix for the disc. For example, exosomal matrilin-3 from urine-derived stem cell exosomes promotes NP cell proliferation and extracellular matrix synthesis by activating TGF-β signaling [163]. Controlled release of TGF-β1 by pullulan microbeads can also lead to an increase in NP cellularity, collagen type II and aggrecan staining intensities, and the Tie^2+^ progenitor cell density in the disc [164]. Notably, the activation of the TGF-β signaling pathway can promote the pro-fibrotic effect of bleomycin on AF cells and BMSCs, which induces rapid fibrosis and height maintenance for IVD. Moreover, bleomycin-induced fibrosis also improves the stress tolerance of the degenerative disc [165].

## 8. Conclusions and Outlook

As mentioned above, there is a complex network among cellular signaling pathways for the IDD process. Stem cells, with regulatory roles in the signaling network, revealed great potential for biological cell-based treatment of IDD. As mentioned above, activation of PI3K/AKT, Shh, and TGF-β signaling pathways, and inhibition of NF-κB and JNK signaling pathways induce IDD remission with stem cell treatment, and the roles of Wnt/β-catenin, ERK1/2, and p38-MAPK pathways in stem-cell-treated IDD remain two-sided. Notably, IVD signaling pathway markers generally precede morphological alterations. A possible approach to improve the clinical practice of IDD-related diseases is to incorporate biomarkers in diagnosis, therapeutic intervention, and prognosis prediction. However, early IDD clinical markers were still lacking in practice. Thus, exploring biomarkers in specific signaling pathways for IDD, as well as stem cells with regulatory effects for these biomarkers, has a high potential value in clinical applications.

## Figures and Tables

**Figure 1 biomedicines-11-02467-f001:**
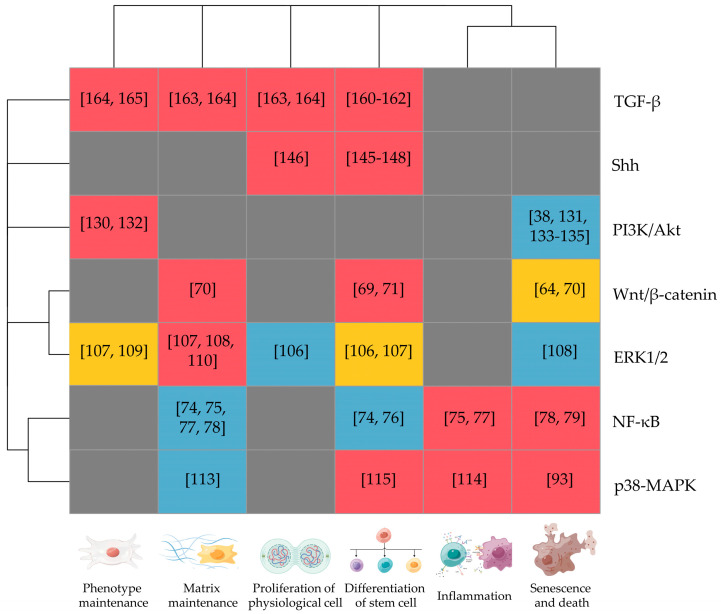
Effects of stem cell treatment and roles of related signaling pathways on IDD progression. Red module: positive relations between the activations of signaling pathway and corresponding biological processes. Blue module: negative relations between the activations of signaling pathway and corresponding biological processes. Yellow module: relations between the activations of signaling pathways and corresponding biological processes varied in different studies. Grey module: lack of relevant evidence. (The figure was created with Figdraw and the OmicStudio tools at https://www.omicstudio.cn on 25 August 2023).

**Table 2 biomedicines-11-02467-t002:** Effects of the MAPK signaling pathway activation for cells in IVD.

Sub-Pathways in MAPK Pathway	Inflammation	Oxidative Stress	Senescence and Death	Proliferation	Phenotype Maintenance	Matrix Maintenance
ERK1/2 signaling pathway	↑[80]	↑[22]	↓[81]↑[22]	↑[82,83,84]	↑[85,86,87]↓[88]	↑[84]↓[54,80]
p38-MAPK signaling pathway	↑[57,89,90,91,92,93,94]	↑[94,95]	↑[56,93,96,97]	↓[92]	↓[92,98]	↓[57]
JNK signaling pathway	↑[59,62,99]	↑[100]	↑[55,60,61,101]	↑[102]	↓[58]	↓[60,62,99]

↑: Promoting effect. ↓: Inhibitory effect.

## Data Availability

Not applicable.

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
