# Peer review of "Signaling Mechanisms of Stem Cell Therapy for Intervertebral Disc Degeneration"

_biomedicines, 2023, doi:10.3390/biomedicines11092467_

Round 1
Reviewer 1 Report
Dear Authors,
Your publication clearly shows the mechanisms responsible for the development of IDD as well as endogenous and exogenous factors that may participate in stem cell therapy for intervertebral disc degeneration. It shows the possibilities and therapeutic prospects as possible pharmacological targets for the development of new drugs.
I propose to develop a table or extend Table 1 to show the signaling pathways that are discussed and what effect they have on IDD progression. Show in this table that, for example, from the point of view of IDD therapy, it is desirable to inhibit the Wnt/β-catenin pathway, while another signaling pathway, such as NF-κB, should be inhibited, etc. This will make the publication much easier to read and understand.
Minor notes:
1. Briefly explain what RBMS3 is and what it does.
2. On line 144, is there supposed to be "inflammation factors", maybe it would be better to use "inflammatory factors"?
Reviewer 2 Report
The paper regarding signaling mechanisms of stem cell therapy for intervertebral disc degeneration is potentially interesting. The review is well presented.
There are some possible issues
Recent studies using single-Cell RNA Sequencing (for example PMID: 35874809) have found that in IVD, several cell types were present, which were chondrocyte 1, chondrocyte 2, chondrocyte 3, chondrocyte 4, chondrocyte 5, endothelial, macrophage, neutrophil, and T cells. It would be relevant to discuss which cell types are more involved in intervertebral disc degeneration.
In 4.2. p38-MAPK signaling pathway in degenerative IVD cells, It would be informative to discuss possible involvements of Extracellular-signal-regulated kinase like ERK5 ( for example PMID: 24857985) as well.
It is suggested to include key diagrams/figures to illustrate various signaling pathways and mechanisms involved in intervertebral disc degeneration.
Reviewer 3 Report
The manuscript titled “Signaling mechanisms of stem cell therapy for intervertebral disc degeneration” summarizes and describes recent advances in IDD-related signaling pathways and related stem cell treatment in IDD. The topic is relevant and deserves consideration. The content of the review is suitable and the work is well-designed. However, where are some recommendations that can help to make the manuscript better.
1. Abstract contains many abbreviations. It is better to avoid abbreviations usage in Abstract, or the authors can define all cases.
2. The aim of the manuscript should be mentioned in the Background. Why the authors decided to make this review? How it will be useful for community?
3. In general, the manuscript is “dry” and needs some figures to improve visual perception
4. In addition to Outlook, Conclusions is also needed to summarize the main important aspect found during review.
the text should be check for typos
Round 2
Reviewer 2 Report
This a revised paper. The authors have addressed questions with improvement.